# OpenReview forum: "The Pitfalls of KV Cache Compression"
_ICLR.cc/2026/Conference — ICLR 2026 Conference Withdrawn Submission_

### Official Review · Reviewer_jNmw · 2025-10-27

**Soundness:** 2
**Presentation:** 2
**Contribution:** 2
**Rating:** 2
**Confidence:** 4

**Summary:**

This paper investigates the often-overlooked side effects of Key-Value (KV) cache compression in Large Language Models (LLMs), arguing that its consequences in realistic, multi-instruction scenarios are poorly understood. The authors demonstrate that performance does not degrade uniformly; rather, compression can cause "selective amnesia”, where certain instructions—particularly system prompts and safety guardrails—are disproportionately affected and silently ignored. This leads to critical failures, such as system prompt leakage. The study identifies that this vulnerability is influenced by the specific compression method, the order of instructions, and an "eviction bias" in many policies. To address this, the paper proposes two practical modifications: manually "whitelisting" critical tokens and implementing a "fair eviction" policy to ensure all instructions are compressed at a more equal rate.

**Strengths:**

- The paper studies the important and practical problem of how KV cache compression, while improving efficiency, can introduce unintended side effects that harm model alignment and safety.
- It moves beyond simple performance metrics to analyze a critical failure mode (system prompt leakage) that is highly relevant for deploying models in production.
- It provides practical recommendations for mitigating this degradation by modifying eviction policies to retain certain important KV cache entries.

**Weaknesses:**

- The core finding that performance degrades with a reduced KV cache budget is a well-known trade-off, and most existing literature on KV cache compression already documents utility degradation as the compression ratio increases.
- The study primarily uses the IFEval dataset, which is not a long-context benchmark. The argument would be significantly more convincing if it demonstrated these instruction-following failures on a benchmark explicitly designed to test long-context capabilities.
- The whitelisting solution relies on manual effort and user intuition to identify critical tokens. This approach is not scalable and cannot be generalized, as it would require bespoke tuning for every new set of system prompts or instructions.
- The proposed "fair eviction" policy raises questions. If eviction impacts all instructions equally, this could also be a negative outcome, as it might degrade critical instructions at the same rate as trivial ones, rather than intelligently prioritizing the most important information. The utility of this fairness seems questionable if it leads to uniform degradation.
- The relative impact of eviction across different layers/heads is not studied. The paper applies its eviction policies globally, but the effects of compression may not be uniform across all layers/heads.

**Questions:**

- How does the "selective amnesia" you observe differ significantly from the known performance/compression trade-offs already established in prior work?

- Could the instruction-following failures observed on IFEval be replicated or potentially magnified on dedicated long-context benchmarks?

- Regarding "fair eviction," could this policy be counter-productive by failing to prioritize critical instructions (like safety guardrails) over trivial ones, leading to a uniform, but still harmful, degradation?

- How could the "whitelisting" approach be scaled or automated? As it stands, doesn't it require manual, prompt-specific tuning that isn't generalizable?

- You note that policies are applied globally. Did you investigate the layer-wise sensitivity to compression? Is it possible that evicting tokens from specific layers (e.g., middle vs. final) has a disproportionate impact on instruction following?

---

> ### Author Response · Authors · 2025-11-27
>
> We would like to thank the reviewer for their suggestions and comments. We shall address questions below. We have also uploaded an updated PDF with the requested changes to the main body of the text and additional experiments (see Appendices). Changes are marked in blue.
>
> > The core finding that performance degrades with a reduced KV cache budget is a well-known trade-off, and most existing literature on KV cache compression already documents utility degradation as the compression ratio increases.
> >
> > How does the "selective amnesia" you observe differ significantly from the known performance/compression trade-offs already established in prior work?
>
> Indeed, single instruction performance degradation is well-known and present in the literature (as discussed in Section 3), but that is not the object of study of our paper. Our paper specifically studies the multiple instruction case (which is present in most system prompts used in the real-world) and we make this distinction in Section 3, arguing that most KV cache eviction methods are optimized to the (less realistic) single instruction case. In fact, we show that popular KV eviction methods break down on the multi-instruction case as a consequence of eviction bias yet can be readily patched with something as simple as whitelist or fair eviction. As far as we know, our contributions, namely (1) that each instruction in a multi-instruction prompt degrades at different rates, (2) the consequences of this difference and (3) the presence of eviction bias within most eviction policies—all highlighted in Pitfalls 3, 4 and 5—, have never been studied before and are novel.
>
> > The study primarily uses the IFEval dataset, which is not a long-context benchmark. The argument would be significantly more convincing if it demonstrated these instruction-following failures on a benchmark explicitly designed to test long-context capabilities.
> >
> > Could the instruction-following failures observed on IFEval be replicated or potentially magnified on dedicated long-context benchmarks?
>
> Our experiments show that eviction bias is about relative keep-rate imbalance, not absolute context length. Therefore, even short-context evaluations expose the underlying failure mode. That being said, we have supplemented additional experiments with the longbench dataset [1], discussed in detail in Appendix G. We replaced IFEval with Longbench’s Trec dataset. Trec provides question classification examples and evaluates the accuracy of the model’s classification on an unseen question. Trec’s in-context learning framework is suitable for our application because our paper investigates the degradation of orthogonal instructions, which in this case is leakage prevention vs question classification. Retrieval-based long-context benchmarks are unsuitable as retrieval and leakage prevention both assess the extent to which the model reveals the system prompt. We also looked into two other Longbench in-context learning datasets: Samsum and Triviaqa. While Trec shows meaningful question classification degradation with compression, the others do not have such a pattern, most likely due to the in-context learning examples not being very important for answering their respective unseen questions. As such, we only consider our results for Trec.
>
> Before we discuss the results, we would like to clarify the motivation behind our experiment: we want to illustrate eviction bias through the differing degradations of orthogonal instructions. To achieve this, we need to choose a system prompt leakage defense template that is not excessively strong or weak. If the defense is too strong, the system prompt is never leaked. If the defense is too weak, the system prompt is leaked even at 0 compression. We discuss experimental choices in Appendix G.
>
> We believe that the figures in Appendix G illustrate the main point of our paper in the long-context setting: when one instruction is unfairly prioritized over the other, bias correction is  beneficial. We talk about this in more detail in Appendix F and in our answer to the next question.
>
> [1] - https://arxiv.org/abs/2308.14508
>
> We continue to answer the other questions in the next comment.

---

> ### Author Response · Authors · 2025-11-27
>
> > The proposed "fair eviction" policy raises questions. If eviction impacts all instructions equally, this could also be a negative outcome, as it might degrade critical instructions at the same rate as trivial ones, rather than intelligently prioritizing the most important information. The utility of this fairness seems questionable if it leads to uniform degradation.
> >
> > Regarding "fair eviction," could this policy be counter-productive by failing to prioritize critical instructions (like safety guardrails) over trivial ones, leading to a uniform, but still harmful, degradation?
>
> We agree that fair eviction may not achieve optimal performance in all cases. An underlying assumption about fair eviction is that instruction blocks are equal in importance and semantic complexity. We recognize that this has not been explicitly stated in the paper, and updated the fair eviction section accordingly (see lines 470-472 in the updated PDF). Furthermore, we would like to emphasize that the purpose of our paper is not to come up with state-of-the-art eviction policies, but to provide evidence for the existence of eviction bias  and its drawbacks, which should be more well understood and studied in future work.
>
> However, we do recognize that this is a very interesting question, and thank the reviewer for this suggestion. We have investigated the trade-off between enforcing fairness and achieving optimal performance, detailing the experiment in Appendix F. At a high level, in the case of two instructions, we interpolate (through a parameter $\lambda$) between the tokens kept per instruction. We use $\lambda$ to enforce token splits between the two instructions $X$ and $Y$ through the equation $\text{keep}_X(\lambda) = \lambda \text{keep}_X(1) + (1 - \lambda)\text{keep}_X(0)$ and $\text{keep}_Y = \lambda \text{keep}_Y(1) + (1 - \lambda)\text{keep}_Y(0)$, where $\text{keep}_X(0)$ (resp. $Y$) is defined to be the (unmodified) number of kept entries of the original eviction policy in $X$ (resp. $Y$), and $\text{keep}_X(1)$ (resp. $Y$) defined as the number of kept entries in $X$ (resp. $Y$) after fair eviction. In other words, $\lambda = 0$ is the same as regular compression and lambda = 1.0 is fair eviction. We plotted the system prompt leakage against the system directive following compression ratios 0.0, 0.1, …, 0.9 for StreamingLLM on the Ifeval dataset. We observe that, on average, fair eviction is 30% more Pareto optimal than default compression, and has the highest optimality. We categorize optimality through Pareto optimality, which means there is no other $\lambda$ that achieves better performance in both leakage and system directive following. From these new experiments, we substantiate our original claim that KV compression is suboptimal for prompts with multiple instructions and that eviction policies need to be made more "fair" w.r.t. eviction bias.
>
> As requested by other reviewers, we also investigated long-context datasets and showed that the same results persisted across SnapKV and Tova. Fair eviction is 25% more optimal than default compression, and has the highest optimality.
>
> Once again, our focus is on proving the existence of eviction bias and providing tangible solutions to correcting the bias. We encourage future research into optimizing this eviction debiasing.
>
> We continue to answer the other questions in the next comment.

---

> ### Author Response · Authors · 2025-11-27
>
> > The whitelisting solution relies on manual effort and user intuition to identify critical tokens. This approach is not scalable and cannot be generalized, as it would require bespoke tuning for every new set of system prompts or instructions.
> >
> > How could the "whitelisting" approach be scaled or automated? As it stands, doesn't it require manual, prompt-specific tuning that isn't generalizable?
>
> In this paper we study offline compression, where the user knows exactly what the prompt to be compressed is. The common setting for offline compression is system prompts (which we study in the paper). The system prompt, which can be quite long, is compressed at the KV cache level in order to avoid unnecessary recomputation while still maintaining the semantics of the prompt. This is in contrast to online compression, where the goal is to compress the (rolling) context while sampling. We recognize that we do not mention that we are specifically studying offline compression, and added a proper explanation to the updated PDF (lines 138 - 156).
>
> Thus, in our offline setting, the user does have a priori knowledge on the prompt by definition, and can use this information to best compress their prompt. Still, one can adapt whitelist and fair eviction in order to either automate this manual step: for whitelist eviction, one can use an LLM to identify what the most important sentences in the prompt are and whitelist them accordingly; for fair eviction, one can select the instruction spans at the sentence level (every sentence should then be fairly evicted) or use an LLM to identify the instruction spans at the semantic level and automatically apply fair eviction this way. A discussion on automation is in Appendix K.
>
> > The relative impact of eviction across different layers/heads is not studied. The paper applies its eviction policies globally, but the effects of compression may not be uniform across all layers/heads.
> >
> > You note that policies are applied globally. Did you investigate the layer-wise sensitivity to compression? Is it possible that evicting tokens from specific layers (e.g., middle vs. final) has a disproportionate impact on instruction following?
>
> Applying compression to every layer is the usual practice in KV cache eviction, and so we sought to emulate the consequences and pitfalls of existing KV cache compression methods during our experiments. However, to answer the reviewer’s question, we did run experiments on how different eviction policies behave at each layer, as shown in Appendix J.
>
> The trends in layer-wise sensitivity to compression differ between compression methods. For example, H2O and SnapKV seem to keep defensive tokens at oscillating rates across layers. Tova prioritizes defensive tokens a lot more in the middle layers. We believe a thorough investigation into why these patterns occur warrants an entirely new paper with its own claims and conclusions, as it would entail a detailed explanation of each of the compression methods and a thorough investigation on the mechanistic side of the dynamics of attention within a compressed KV cache setting.

---

### Official Review · Reviewer_ea6q · 2025-10-31

**Soundness:** 3
**Presentation:** 3
**Contribution:** 3
**Rating:** 6
**Confidence:** 3

**Summary:**

This paper raises awareness on the non-uniform behavior/performance of the KV cache compression algorithms. In particular, settings with multi-instruction prompting seem to be more affected. In addition, the paper conducts experiments that show the performance is influenced by the cache replacement algorithms used. As a use case study, the paper focuses on system prompt leakage (e.g., defensive prompts). They show several factors affecting the leakage, such as instruction order, compression algorithm and cache eviction methods. The paper also proposes fair eviction policies that enforce that the fraction of tokens kept from different partitions of the context/prompt is similar.

**Strengths:**

Extensive experiments to showcase the performance of KV caches in different scenarios

**Weaknesses:**

The methods proposed may require a priori knowledge on prompt structure and/or a prompt structure that is static throughout the lifetime of the LLM, which does not seem as a realistic setting

**Questions:**

Does the whitelisting and fair eviction require awareness of the prompt structure? What if the prompt structure is not fixed for the usage of an LLM and is dynamic, depending on the use cases (which is more realistic)?

---

> ### Author Response · Authors · 2025-11-27
>
> We would like to thank the reviewer for their suggestions and comments. We shall address questions below. We have also uploaded an updated PDF with the requested changes to the main body of the text and additional experiments (see Appendices). Changes are marked in blue.
>
> > The methods proposed may require a priori knowledge on prompt structure and/or a prompt structure that is static throughout the lifetime of the LLM, which does not seem as a realistic setting
> >
> > Does the whitelisting and fair eviction require awareness of the prompt structure? What if the prompt structure is not fixed for the usage of an LLM and is dynamic, depending on the use cases (which is more realistic)?
>
> In this paper we study offline compression, where the user knows exactly what the prompt to be compressed is. The common setting for offline compression is system prompts (which we study in the paper). The system prompt, which can be quite long, is compressed at the KV cache level in order to avoid unnecessary recomputation while still maintaining the semantics of the prompt. This is in contrast to online compression, where the goal is to compress the (rolling) context while sampling. We recognize that we do not mention that we are specifically studying offline compression, and have added a proper explanation to the updated PDF (lines 138 - 156).
>
> Thus, in our offline setting, the user does have a priori knowledge on the prompt by definition, and can use this information to best compress their prompt. Still, one can adapt whitelist and fair eviction in order to automate this manual step: for whitelist eviction, one can use an LLM to identify what the most important sentences in the prompt are and whitelist them accordingly; for fair eviction, one can select the instruction spans at the sentence level (every sentence should then be fairly evicted) or use an LLM to identify the instruction spans at the semantic level and automatically apply fair eviction this way. Automation is described in more detail in Appendix K.

---

### Official Review · Reviewer_mxtm · 2025-11-01

**Soundness:** 3
**Presentation:** 2
**Contribution:** 2
**Rating:** 4
**Confidence:** 4

**Summary:**

This paper studies KV Cache compression and its effects on performance, with a focus on multi-instruction prompts. The paper performs a study which shows that in multi-instruction settings, some instructions degrade more rapidly with compression than others. As a practical example to this, the paper shows that system prompting as multiple instructions can lead to leakage of private and proprietary information. The paper proposes simple changes to KV cache eviction policies that reduce the impact of imbalance of instruction degradation to achieve more balanced and predictable performance in the presence of compression.

**Strengths:**

+ The paper highlights important problems such as prompt leakage due to KV cache compression
+ The paper does a good job of isolating the different problems with KV cache compression, such as eviction bias, and order of instruction, etc.

**Weaknesses:**

- While the paper does show different experiments that expose the different aspects of KV cache compression, most of the insights were expected and unsurprising
- It seems that token whitelisting is quite straightforward and not very hard, especially to be used for system prompts, and which can solve this problem. It would be good if there is discussion about some policy that cannot trivially include whitelisting of tokens, and where fair eviction is actually helpful due to its simplicity.
- While the fair eviction solution proposed by the paper does well on the benchmark that the paper runs, it remains unclear whether the solution is good as a generic solution for multi-instruction prompting

**Questions:**

- Please try to answer as many questions as possible from the weakness section.

- When it comes to the fair eviction policy that you propose, is the policy fully automatic? I imagine that even in your policy, somehow you have to partition the prompt into its corresponding instructions to be able to apply fair eviction? How do you do this automatically?

- Would blindly removing an equal number of tokens from each instruction be practical and accurate in realistic scenarios? Intuitively, it seems like different instructions may have different importance and you might want a different ratio in practice, which is much harder to achieve. Could you convince that fair eviction is a generic and always better policy for KV cache eviction?

---

> ### Author Response · Authors · 2025-11-27
>
> We would like to thank the reviewer for their suggestions and comments. We shall address questions below. We have also uploaded an updated PDF with the requested changes to the main body of the text and additional experiments (see Appendices). Changes are marked in blue.
>
> > While the paper does show different experiments that expose the different aspects of KV cache compression, most of the insights were expected and unsurprising
>
> We believe that our results are novel and, although unsurprising in hindsight, very relevant to the KV cache eviction community. As we discuss in Section 3, degradation is obvious and unsurprising for the single instruction case (that has been extensively studied in the literature); our contribution touches on the (not so obvious) multiple instruction case. As far as we know, our contributions, namely (1) that each instruction (in a multi-instruction prompt) degrades at different rates, (2) the consequences of this difference and (3) the presence of eviction bias within most eviction policies—all highlighted in Pitfalls 3, 4 and 5—, have never been studied before and are novel. Further, we show that although popular KV eviction methods break down on the multi-instruction case as a consequence of eviction bias, they can be readily patched with something as simple as whitelist or fair eviction.
>
> > It seems that token whitelisting is quite straightforward and not very hard, especially to be used for system prompts, and which can solve this problem. It would be good if there is discussion about some policy that cannot trivially include whitelisting of tokens, and where fair eviction is actually helpful due to its simplicity
>
> We believe that whitelist eviction and fair eviction tackle two different problems: the former presents evidence for Pitfall 6, showing that the choice of which entries to evict is important; the latter shows the existence of eviction bias as presented in Pitfall 5. These two problems can coexist.
>
> Having said that, even though token whitelisting is simple conceptually, it still heavily relies on manual effort and user intuition, as stated in lines 468-470. The whitelisting examples shown in the paper are actually good examples of non-trivial whitelisting of tokens. We iterated over many combinations of whitelisted tokens, including keyphrases and keywords, and eventually settled with what was written in Appendix C. In contrast, fair eviction does achieve similar results in terms of performance yet requires minimal effort, as it is mostly automatic modulo deciding where the instruction spans are (though they can be easily identified).
>
> > While the fair eviction solution proposed by the paper does well on the benchmark that the paper runs, it remains unclear whether the solution is good as a generic solution for multi-instruction prompting
>
> We would like to emphasize that the main point of our paper is to show evidence of the problems and pitfalls in KV cache eviction, which are backed up by our experiments on the IFEval dataset, meaning that we do not think of fair eviction as a generic solution, but rather as a tool to highlight the fact that eviction bias (Pitfall 5) exists. However, we did conduct additional experiments with the longbench dataset [1], discussed in detail in Appendix G. We discuss fair eviction’s justification in Appendix F.
>
> [1] - https://arxiv.org/abs/2308.14508
>
> We continue to answer the other questions in the next comment.

---

> ### Author Response · Authors · 2025-11-27
>
> > When it comes to the fair eviction policy that you propose, is the policy fully automatic? I imagine that even in your policy, somehow you have to partition the prompt into its corresponding instructions to be able to apply fair eviction? How do you do this automatically?
>
> An important note to clarify here is that in this paper we study offline compression, meaning that the user knows what the prompt contains and as such can use their prior knowledge in order to manually apply whitelist or fair eviction. Still, both whitelist and fair eviction can be easily automated: for whitelist, one can utilize an LLM to highlight what the most important sentences of the (to be compressed) prompt are and use that as the whitelist; for fair eviction, one can assume that each sentence in the prompt is an instruction span and automatically apply fair eviction this way, or alternatively use an LLM to delineate more semantically distinct instruction spans. We discuss this automation in more detail in Appendix K.
>
> Although both whitelist and fair eviction can be automated, the point of our paper is to raise awareness on the many pitfalls of KV cache eviction, with whitelist eviction backing up the claim of Pitfall 6 and fair eviction showing the importance of Pitfall 5.
> Pitfall 6 and whitelist eviction show that there is a lot to improve with regards to choosing which entries to evict: a human can beat a sophisticated algorithm that at every layer automatically chooses which entries to evict according to some complicated policy by simply choosing a single semantically important sentence.
>
> Pitfall 5 and fair eviction show that eviction bias is a phenomenon that was not previously known and can cause issues to performance; even a simple change in the direction of reducing this bias like fair eviction can drastically improve performance for multi-instruction prompts.
>
> > Would blindly removing an equal number of tokens from each instruction be practical and accurate in realistic scenarios? Intuitively, it seems like different instructions may have different importance and you might want a different ratio in practice, which is much harder to achieve. Could you convince that fair eviction is a generic and always better policy for KV cache eviction?
>
> We agree that fair eviction may not achieve optimal performance in all cases. An underlying assumption about fair eviction is that instruction blocks are equal in importance and semantic complexity. We recognize that this has not been explicitly stated in the paper, and updated the fair eviction section accordingly (see lines 470-472 in the updated PDF). Furthermore, we would like to emphasize that the purpose of our paper is not to come up with state-of-the-art eviction policies, but to provide evidence for the existence of eviction bias and its drawbacks, which should be more well understood and studied in future work.
>
> However, we do recognize that this is a very interesting question, and thank the reviewer for this suggestion. We have investigated the trade-off between enforcing fairness and achieving optimal performance, detailing the experiment in Appendix F. At a high level, in the case of two instructions, we interpolate (through a parameter $\lambda$) between the tokens kept per instruction. We use $\lambda$ to enforce token splits between the two instructions $X$ and $Y$ through the equation $\text{keep}_X(\lambda) = \lambda \text{keep}_X(1) + (1 - \lambda)\text{keep}_X(0)$ and $\text{keep}_Y = \lambda \text{keep}_Y(1) + (1 - \lambda)\text{keep}_Y(0)$, where $\text{keep}_X(0)$ (resp. $Y$) is defined to be the (unmodified) number of kept entries of the original eviction policy in $X$ (resp. $Y$), and $\text{keep}_X(1)$ (resp. $Y$) defined as the number of kept entries in $X$ (resp. $Y$) after fair eviction. In other words, $\lambda = 0$ is the same as regular compression and lambda = 1.0 is fair eviction. We plotted the system prompt leakage against the system directive following compression ratios 0.0, …, 0.9 for StreamingLLM on the Ifeval dataset. We observe that, on average, fair eviction is 30% more Pareto optimal than default compression, and has the highest optimality. We categorize optimality through Pareto optimality, which means there is no other $\lambda$ that achieves better performance in both leakage and system directive following. From these new experiments, we substantiate our original claim that KV compression is suboptimal for prompts with multiple instructions and that eviction policies need to be made more fair w.r.t. eviction bias.
>
> As requested by other reviewers, we also investigated long-context datasets and showed that the same results persisted across SnapKV and Tova. Fair eviction is 25% more optimal than default compression, and has the highest optimality.
> Once again, our focus is on proving the existence of eviction bias and providing tangible solutions to correcting the bias. We encourage future research into optimizing this eviction debiasing.

---

### Official Review · Reviewer_KrPd · 2025-11-01

**Soundness:** 2
**Presentation:** 3
**Contribution:** 2
**Rating:** 4
**Confidence:** 5

**Summary:**

This paper intends to investigate the failure modes of KV cache compression, arguing that performance degrades non-uniformly, causing LLMs to silently ignore parts of a prompt, a phenomenon the authors term "selective amnesia."  Using system prompt leakage as a case study for this security vulnerability, the work systematically analyzes several eviction-based compression policies. It identifies key contributing factors, including the compression method, instruction order, and a newly defined "eviction bias," where certain instructions are disproportionately targeted for eviction. The paper's main contributions are to identify and characterize these pitfalls and to propose and evaluate two simple mitigation strategies ("whitelisting" and "fair eviction") to counteract this bias.

**Strengths:**

Novel and Important Problem Framing: The paper's primary strength is reframing KV cache compression from a simple efficiency trade-off to a critical issue of model reliability and security. The identification of "selective amnesia" and its connection to system prompt leakage is an insightful and timely contribution, as such silent failures are a major concern for deploying LLMs in production.

Systematic Empirical Analysis: The claims are well-supported by a rigorous set of experiments across multiple models and five different eviction policies. The analysis of instruction ordering and "eviction bias" provides convincing evidence that the observed failures are systematic rather than random.

Clear Presentation: The paper is well-written and clearly structured. The core concepts are explained effectively, and the use of visualizations (e.g., Figure 1) makes the findings accessible and impactful.

**Weaknesses:**

Overclaiming of Scope: The paper makes broad claims about "KV cache compression" but exclusively evaluates eviction-based policies. It fails to consider other major paradigms like quantization or merging , which may not exhibit the same failure modes. This mismatch between the claims and the evidence is a significant limitation.

Superficial Solutions and Incomplete Evaluation: The proposed solutions ("whitelisting" and "fair eviction") are simple heuristics that lack novelty. Whitelisting is not a scalable, automated solution, while "fair eviction" is a blunt instrument that may be suboptimal. The evaluation of these methods is also incomplete, as it omits performance at high compression ratios where their trade-offs would be most apparent.

Lack of Mechanistic Insight: The paper effectively describes what happens (e.g., eviction bias occurs) but provides little explanation for why it happens. A deeper analysis connecting the observed phenomena to the underlying mechanics of the attention mechanism and the specific logic of each eviction policy is missing.

Lack of Tradeoff Analysis: Will token recomputation mitigate the loss of accuracy and fairness? How does the proposed solutions affect inference speed?

**Questions:**

Regarding Scope: Your paper makes broad claims about "KV cache compression" but only evaluates eviction-based methods. Given that other paradigms like quantization and merging exist (which may not suffer from "selective amnesia" in the same way), would you consider reframing the paper to be more precisely about the "Pitfalls of KV Cache Eviction Policies"? This would make your claims more strongly supported by the evidence provided.

Regarding "Fair Eviction": Your "fair eviction" policy enforces a uniform retention rate across predefined instruction blocks. Have you considered scenarios where an "unfair" or biased eviction might actually be optimal for a given task (e.g., if one instruction is highly redundant or less semantically complex)? Could you discuss the potential trade-off between enforcing fairness and achieving optimal performance?

Regarding High Compression Ratios: In Figures 9 and 10, the performance of your proposed solutions is not shown for compression ratios above approximately 0.6. Could you provide these results and discuss how your methods perform in the high-compression regime? This is particularly relevant given that Figure 5 shows the baseline leakage problem naturally starts to decrease at very high compression ratios.

Regarding Mechanistic Causes: Your work clearly demonstrates that eviction bias occurs. Could you provide a deeper analysis of why it occurs for the different policies you tested? For instance, for position-based methods like StreamingLLM, is the bias simply an artifact of where the defense prompt is located relative to the attention sink and recent token window? For attention-based methods, what properties of the defense prompt's tokens (e.g., lower norm, different attention patterns) lead to them receiving lower cumulative importance scores?

Regarding the tradeoff: Will token recomputation mitigate the loss of accuracy and fairness? How does the proposed solutions affect inference speed?

---

> ### Author Response · Authors · 2025-11-27
>
> We would like to thank the reviewer for their suggestions and comments. We shall address questions below. We have also uploaded an updated PDF with the requested changes to the main body of the text and additional experiments (see Appendices). Changes are marked in blue.
>
> > Regarding Scope: Your paper makes broad claims about "KV cache compression" but only evaluates eviction-based methods. Given that other paradigms like quantization and merging exist (which may not suffer from "selective amnesia" in the same way), would you consider reframing the paper to be more precisely about the "Pitfalls of KV Cache Eviction Policies"? This would make your claims more strongly supported by the evidence provided.
>
> As far as we know, the term “KV cache compression” is often used within the community to mean KV cache eviction-based methods. In fact, we point to the following (state-of-the-art) literature [1,2,3,4,5] where all of them refer to their (eviction-based) methods as “KV cache compression” methods. Having said that, we do not feel strongly about changing from compression to eviction to clarify our claims if other reviewers also agree that the change is needed.
>
> [1] - https://arxiv.org/pdf/2401.06104
>
> [2] - https://arxiv.org/pdf/2404.14469
>
> [3] - https://arxiv.org/pdf/2406.11430
>
> [4] - https://arxiv.org/pdf/2306.14048
>
> [5] - https://github.com/NVIDIA/kvpress
>
> > Regarding "Fair Eviction": Your "fair eviction" policy enforces a uniform retention rate across predefined instruction blocks. Have you considered scenarios where an "unfair" or biased eviction might actually be optimal for a given task (e.g., if one instruction is highly redundant or less semantically complex)? Could you discuss the potential trade-off between enforcing fairness and achieving optimal performance?
>
> We agree that fair eviction may not achieve optimal performance in all cases. An underlying assumption about fair eviction is that instruction blocks are equal in importance and semantic complexity. We recognize that this has not been explicitly stated in the paper, and updated the "fair eviction" section accordingly (see lines 470-472 in the updated PDF). Furthermore, we would like to emphasize that the purpose of our paper is not to come up with state-of-the-art eviction policies, but to provide evidence for the existence of eviction bias  and its drawbacks, which should be more well understood and studied in future work.
>
> However, we do recognize that this is a very interesting question, and thank the reviewer for this suggestion. We have investigated the trade-off between enforcing fairness and achieving optimal performance, detailing the experiment in Appendix F. At a high level, in the case of two instructions, we interpolate (through a parameter $\lambda$) between the tokens kept per instruction. We use $\lambda$ to enforce token splits between the two instructions $X$ and $Y$ through the equations $\text{keep}_X(\lambda) = \lambda\cdot\text{keep}_X(1) + (1 - \lambda)\cdot\text{keep}_X(0)$ and $\text{keep}_Y = \lambda\cdot\text{keep}_Y(1) + (1 - \lambda)\cdot\text{keep}_Y(0)$, where $\text{keep}_X(0)$ (resp. $Y$) is defined to be the (unmodified) number of kept entries of the original eviction policy in $X$ (resp. $Y$), and $\text{keep}_X(1)$ (resp. $Y$) defined as the number of kept entries in $X$ (resp. $Y$) after fair eviction. In other words, $\lambda = 0$ is the same as regular compression and lambda = 1.0 is fair eviction. We plotted the system prompt leakage against the system directive following compression ratios 0.0, 0.1, …, 0.9 for StreamingLLM on the Ifeval dataset. We observe that, on average, fair eviction is 30% more Pareto optimal than default compression, and has the highest optimality. We categorize optimality through Pareto optimality, which means there is no other $\lambda$ that achieves better performance in both leakage and system directive following. From these new experiments, we substantiate our original claim that KV compression is suboptimal for prompts with multiple instructions and that eviction policies need to be made more "fair" w.r.t. eviction bias.
>
> As requested by other reviewers, we also investigated long-context datasets and showed that the same results persisted across SnapKV and Tova. Fair eviction is 25% more optimal than default compression, and has the highest optimality.
> Once again, our focus is on proving the existence of eviction bias and providing tangible solutions to correcting the bias. We encourage future research into optimizing this eviction debiasing.
>
> We continue to answer the other questions in the next comment.

---

> ### Author Response · Authors · 2025-11-27
>
> > Regarding High Compression Ratios: In Figures 9 and 10, the performance of your proposed solutions is not shown for compression ratios above approximately 0.6. Could you provide these results and discuss how your methods perform in the high-compression regime? This is particularly relevant given that Figure 5 shows the baseline leakage problem naturally starts to decrease at very high compression ratios.
>
> There are two reasons why compression ratios 0.8 and 0.9 in Figures 9 and 10 have been omitted.
>
> The first reason is that, as pointed out in Section 4 (lines 317-320), the information at compression ratios 0.8 and 0.9 is almost completely lost, meaning that the LLM distribution is closer to the prior and the generated responses from the LLM lose any context from the (compressed) prompts. We do not omit these compression ratios for earlier figures since this explanation is part of the observations made in Section 4.
>
> The second reason is that, for Figure 9 compression ratios 0.8 and 0.9 cannot be computed since the number of kept tokens would then be less than the number of whitelisted tokens. We thus wanted to keep the same x-axis for both whitelist and fair evictions for better visualization.
>
> Having said that, we appreciate the reviewers comment and shall update the PDF with compression ratios 0.8 and 0.9 for Figure 10.
>
> > Regarding Mechanistic Causes: Your work clearly demonstrates that eviction bias occurs. Could you provide a deeper analysis of why it occurs for the different policies you tested? For instance, for position-based methods like StreamingLLM, is the bias simply an artifact of where the defense prompt is located relative to the attention sink and recent token window? For attention-based methods, what properties of the defense prompt's tokens (e.g., lower norm, different attention patterns) lead to them receiving lower cumulative importance scores?
>
> Yes, StreamingLLM’s eviction bias is an artifact of where the defense prompt is located relative to the system directive instructions. Because StreamingLLM preserves the first 4 tokens and the recent token window, if the defense comes first, more of the defense tokens are evicted. While this flips cleanly if we swap the ordering of the instructions, eviction bias in other compression methods and their associated degradation patterns do not, as stated in lines 329 - 331.
>
> A complete explanation of eviction bias in attention-based and embedding-based methods is hard to achieve as even within the same category, different methods have different eviction bias patterns. For example, when the defense comes before the system directive, H2O and SnapKV evict more defense instructions while Tova evicts more system directive instructions. They all utilize attention for deciding which tokens to keep. The underlying reason as to why eviction bias occurs for each method would require a thorough analysis into each of the methods and their particularities, which we believe that, although useful, would be a paper in itself. That being said, we agree that a high-level hypothesis would be helpful, and have added a detailed discussion to the PDF (Appendix H).
>
> We also investigated the eviction bias per layer in our new long context experiments, showcased in Appendix J. The patterns differ between layers and compression methods, but a thorough analysis as to why that occurs requires a more in-depth investigation that we also believe is beyond the scope of this paper.
>
> > Regarding the tradeoff: Will token recomputation mitigate the loss of accuracy and fairness? How does the proposed solutions affect inference speed?
>
> We are unsure what token recomputation means in this context. Here, we are studying the effect of offline compression, where the user compresses a fixed prompt as a preprocessing step, and then generates text conditioned on the compressed prompt. Therefore, there is no change in inference speed regardless of eviction policy.
>
> With respect to the compression procedure itself—i.e. the preprocessing step where the fixed prompt is compressed—neither whitelist nor fair eviction fundamentally change how the eviction is chosen. Whitelisting only prevents some tokens from being evicted, while fair eviction merely applies the chosen compression method to each instruction span independently.
> There is negligible overhead for whitelisting at compression time: the only additional computation needed is to exclude the whitelisted tokens from the list of tokens to be evicted according to the used compression method. This is a negligible overhead as Appendix I shows.
>
> For fair eviction, the added computation is described in Appendix E: the only change is on identifying each instruction span, computing the number of tokens allowed to be evicted for each span and then running the eviction policy on each span. As Appendix I shows, there is negligible overhead on runtime.

---

### Note · Authors · 2026-01-06

I have read and agree with the venue's withdrawal policy on behalf of myself and my co-authors.